# Effects of Bamboo Forest Type and Density on the Growth of *Bletilla striata* and Root Endophytic Fungi

**Hao Fu** [1,2], **Zhilin Song** [2], **Shanmin Li** [1], **Siren Lan** [2], **Xinhua Zeng** [1,*] **and Weichang Huang** [1,2,*]

1  Eastern China Conservation Centre for Wild Endangered Plant Resources, Shanghai Chenshan Botanical Garden, Shanghai 201602, China; fuhaoxsxz@163.com (H.F.); lishanmin2022@163.com (S.L.)
2  College of Art and Landscape Architecture, Fujian Agriculture and Forestry University, Fuzhou 350002, China; songzlinlin@163.com (Z.S.); lsr9636@163.com (S.L.)
*  Correspondence: zengxinhua@csnbgsh.cn (X.Z.); huangweichang@csnbgsh.cn (W.H.)

**Abstract:** *Bletilla striata* is a terrestrial orchid with high ornamental and medicinal values that is widely interplanted in bamboo forests. However, little is known about the effects of bamboo forest type and density on the growth of *B. striata* and its symbiotic relationship with root endophytic fungi. In this study, the growth state of *B. striata*, the community composition and diversity of its root endophytic fungal, and the fungal nutritional function were investigated in *Phyllostachys edulis*, *P. iridescens* and *P. glauca* forests with three densities. We found that the type and density of the bamboo forest had significant effects on the growth of *B. striata*, with the biomass, leaf width, root quantity and width being the highest in the low-density of the *P. edulis* forest. The community composition and abundance of root endophytic fungi in *B. striata* varied among different bamboo forests and densities, with *P. edulis* and *P. iridescens* forests dominated by Basidiomycota and *Serendipita*, while *P. glauca* prevailed by Ascomycota and *Dactylonectria*. The trophic modes of root endophytic fungi were also affected by forest types and densities. The abundance of symbiotroph fungi was the highest in *P. edulis* and *P. iridescens* forests and greatly varied with density gradient, and saprotrophic fungi comprised the highest proportion in the *Ph. glauca* forest. These results provide basic data for further research and the configuration between bamboo species and terrestrial orchids.

**Keywords:** *Bletilla striata*; compound system of bamboo and orchid; endophytic fungi; fungal diversity; symbiotic relationship



## 1. Introduction

As an artificial ecosystem that considers ecological, social and economic benefits, the agroforestry system is of great significance for ecological environmental protection and sustainable development of agroforestry [1]. Agroforestry systems are conducive to optimizing the rational use of resources and space by plants, improving system biodiversity and the internal ecological environment of the system (e.g., light, heat, water, soil, air, fertilizer, microorganism) [2,3]. Because of their advantages of diverse output and income balance, agroforestry systems are widely adopted in many parts of the world, especially in developing countries, and play an important role in the national economy and in people's livelihood [4].

Agroforestry systems involve a wide range of woodland types, with bamboo forests being one of the most important types. Bamboo is one of the most important species in terrestrial forest ecosystems that are widely distributed in tropical and subtropical regions, being the second largest forest in the world [5,6]. Bamboo forests have important economic, ecological and social values [7]. Compared with single crop systems, an agroforestry system based on bamboo forests has higher net primary productivity, which has attracted wide international attention especially in tropical regions [8,9]. A bamboo forest composite system has many modes, mainly including bamboo-grass, bamboo-medicinal plants, bamboo-fungus, bamboo cultivation and forest tourism [7,10], of which bamboo-medicinal herbs are

one of the most popular modes at this stage. Researchers have found that there are a variety of medicinal plants suitable for planting in bamboo forests, such as *Polygonatum sibiricum*, *Lophatherum gracile*, *Sarcandra glabra*, *Bletilla striata* and *Polygonatum odoratum* [11], and the impact of bamboo forest composite systems on medicinal plants varies among different stand types, bamboo forest density and habitats [12,13]. Huang et al., reported that canopy density of 0.4~0.6 is the most effective stand density for polysaccharide accumulation of *Polygonatum cyrtonema* [14]. Zhou found that the understory habitat had a significant impact on the growth traits of *Polygonatum cyrtonema*, and the Moso bamboo forest was the best interplanting site [13]. Feng found that different medicinal plants have different density requirements for bamboo forests, with *Polygonatum cyrtonema* adapting to all density ranges, while *Curculigo orchioides* prefers low density and *Paris polyphylla* medium densities [15]. However, at present, research on composite systems of bamboo-medicinal plants have mainly focused on their economic and social value, and thus, little is known about their ecological value, especially soil physical and chemical properties, plant landscape and the interaction between plant species in bamboo forests.

*Bletilla striata* (Thunb. ex A. Murray) Rchb. f. is a perennial herb mainly distributed across China, the Korean Peninsula and Japan, and it belongs to the genus *Bletilla* in the tribe Arethuseae, Orchidaceae [16]. Owing to its high ornamental and medicinal value, *B. striata* is widely used as an ornamental plant and as traditional medicinal material in Asia [17]. *Bletilla striata* is a typical mycorrhizal plant which relies on root endophytic fungi (both mycorrhizal and non-mycorrhizal fungi) throughout its life cycle, especially during seed germination and seedling recruitment periods [18,19]. Researchers have found that root endophytic fungi can not only transport carbohydrates and break down cellulose in the matrix but can also directly provide nutrients and hormones (e.g., amino acids, gibberellins and jasmonate) for plant growth [20,21]. In addition, root endophytic fungi were found to promote the absorption of macronutrient elements and micronutrient elements by plants [22,23] and facilitate the production of metabolites, including antibiotics, phenolic compounds, peroxidase and hydrolase, thus enhancing disease resistance and stress tolerance in orchids [24,25]. Mycorrhizal partners can also influence orchid distributions and determine which habitats allow orchid growth and what environmental factors are critical for orchid recruitment [26].

As a typical plant with a high potential for compound cultivation, *B. striata* has been widely interplanted with bamboo forests in East China. The growth of *B. striata* and its relationship with root endophytic fungi are greatly influenced by many factors such as temperature, light intensity, nutrients, humidity and soil pH, which are strongly influenced by bamboo forest type and density. Zeng et al. found that there were significant temporal variations in the diversity and community composition of root endophytic fungi of *B. striata* [27]. Ma et al. found that nitrogen had a strong influence on the growth and polysaccharides accumulation of *B. striata*, and ammonium-nitrate mixed nitrogen at the concentration of 15 mmol·L$^{-1}$ had the best results [28]. By adjusting the density of bamboo forests in time, the understory can meet various demands of light radiation [12]. Meanwhile, the diurnal and seasonal changes of temperature, moisture and light conditions varied among different bamboo forests and stand densities [13]. Currently, there have been some studies on the cultivation of *B. striata* under bamboo forests. However, most of these studies focused on the water and soil conservation function, land use capability and production efficiency of bamboo forests [29,30]. Few studies have emphasized the landscape effect and root endophytic microbial community of *B. striata* under different bamboo species and its densities.

In this study, we selected three common types of bamboo (*Phyllostachys edulis* (Carr.) J.Houz., *Phyllostachys iridescens* C. Y. Yao et S. Y. Chen and *Phyllostachys glauca* McClure) in Shanghai, China, intercropped with *B. striata* to study the effects of different types of bamboo and bamboo density on the growth of *B. striata* and the community composition of root endophytic fungi. Our main objectives are to answer the following questions: (1) What are the mycorrhizal partners of *B. striata* in a bamboo forest? (2) Does the com-

munity composition of root endophytic fungi vary among different bamboo forest types or different stand densities? (3) What are the most suitable types of bamboo forest and planting density for the growth of *B. striata* and its mycorrhizal association. Our study will provide data and research ideas for the plant configuration under bamboo forests and the application of orchids under the forests.

## 2. Materials and Methods

### 2.1. Overview of the Study Area

The research site was located in Shanghai Chenshan Botanical Garden, which is in the north subtropical monsoon humid climate, with an annual average temperature of 15.6 °C, annual sunshine duration of 1817 h and precipitation of 1313 mm [31]. The bamboo garden was constructed north of the botanical garden in 2016 and covered an area of approximately 3 hm$^2$. There are more than 70 bamboo species, and *Phyllostachys edulis*, *P. iridescens* and *P. glauca* were chosen as our experiment materials.

### 2.2. The Sample Set

Bamboo forests of *Phyllostachys edulis*, *P. iridescens* and *P. glauca* were divided into high-, middle- and low-density bamboo forests. The densities of the *P. edulis* forest were 14,000, 8500 and 4500 plants/hm$^2$, respectively; the densities of the *P. iridescens* forest were 30,000, 22,000 and 15,000 plants/hm$^2$, respectively, and the densities of the *P. glauca* forest were 60,000, 45,000 and 30,000 plants/hm$^2$, respectively. In February 2021, three 4 × 4 m plots were established in each forest density. Each plot was repeated three times, which resulted in a total of 27 plots. One-year tissue culture plantlets of *B. striata* which derived from the same batch and suffered the same hardening–seedling process were used in this study. *B. striata* was planted at 30 × 30 cm spacing in each plot in early March 2021.

### 2.3. Sample Collection and Processing

Root and soil samples of *B. striata* were collected at the end of July 2021. Five healthy *B. striata* plants were randomly selected from each plot. Two vegetative roots were selected from each plant, and the root samples from five plants were mixed as one root sample. The soil on the surface of *B. striata* root samples was washed first with clean water and then with sterile water for 30 s in an ultra-clean workbench. The samples were then washed in 75% alcohol for 30 s, soaked in 3.5% NaClO for 4 min, and washed three times with sterile water. The treated *B. striata* roots were placed in a clean plastic bag and stored at −80 °C for high-throughput sequencing. Simultaneously, the soil samples around the root were collected, and the fresh soil samples were separated as soon as possible and preserved at −20 °C. The rest of the soil was ground after natural air drying and screened through a 60-mesh sieve to determine soil physical and chemical properties together with fresh soil samples.

In mid-October 2021, 10 healthy *B. striata* plants were randomly selected from each plot and taken back to the laboratory for washing. After washing, the plant height, leaf length, leaf width and root length were measured using a measuring tape. Vernier calipers were used to measure the stem thickness, root width and other indicators. The fresh weight of *B. striata* was measured on an electronic balance. Finally, the *B. striata* plants were heated at 105 °C for 20 min and dried to a constant weight at 65 °C. The dry weight was then measured.

### 2.4. Environmental Factor Index Measurement

Soil pH was measured using the potentiometric method. Electrical conductivity (EC) was measured using the extraction and drying method. Total nitrogen (TN), total phosphorus (TP) and total potassium (TK) were determined using the Kjeldahl method, the molybdenum–antimony resistance colorimetric method and NaOH melting flame spectrophotometry, respectively. Ammonium nitrogen (AN) and nitrate nitrogen (NN) were determined using potassium chloride-indophenol blue colorimetry and ultraviolet

spectrophotometry, respectively. Available phosphorus (AP) and available potassium (AK) were determined by sodium bicarbonateHCl extraction and ammonium acetate extraction—flame spectrophotometry, respectively. Organic matter (OM) and microbial carbon (MBC) were determined using potassium dichromate external heating and chloroform fumigation methods, respectively. The moisture content (MC) and soil bulk density (SBD) were measured using the drying and ring knife methods, respectively [32]. Light intensity (LI) in the bamboo forests was measured using a Xima-AS823 illuminance meter (Shenzhen, China) in the growing season of *B. striata* (March to August). Five points were measured in each plot, and the average value was determined six times a month.

### 2.5. High-Throughput Sequencing

An E.Z.N.A.® soil DNA kit (Omega Bio-Tek, Norcross, GA, USA) was used to extract the total community DNA, and 1% agarose gel electrophoresis was used to assess the quality of the DNA. The nuclear ribosomal internal transcribed spacer-1 (ITS-1) region was amplified with the primers ITS1F (5′-CTTGGTCATTTAGagGaAGTAA-3′) and ITS2R (5′-gCTGCGTTCTTCATCGATG-3). The amplification process included: pre-denaturation at 95 °C for 3 min, denaturation at 95 °C for 30 s, annealing at 55 °C for 30 s and extension at 72 °C for 45 s, 35 cycles. It was stably extended at 72 °C for 10 min and finally stored at 10 °C. The PCR reaction system was as follows: 5× TransStart FastPfu Buffer 2 μL, 2.5 mmol/L dNTPs 2 μL, 5 μmol/L upstream primer 0.8 μL, 5 μmol/L downstream primer 0.8 μL, TransStart FastPfu DNA polymerase 0.2 μL and template DNA10 ng, fill to 20 μL. The PCR product was extracted from 2% agarose gel and purified using the AxyPrep DNA Gel Extraction Kit (Axygen Biosciences, Union City, CA, USA) according to manufacturer's instructions and quantified using a Quantus™ Fluorometer (Promega, Madison, WI, USA).

Purified amplicons were pooled in equimolar and paired-end sequenced (2 × 300) on an Illumina MiSeq platform (Illumina, San Diego, CA, USA) according to the standard protocols from Majorbio Bio-Pharm Technology Co. Ltd. (Shanghai, China). Raw fastq files were demultiplexed and quality filtered with Trimmomatic according to the following criteria: (i) The 300 bp reads were truncated at any site receiving an average quality score of <20 over a 50 bp sliding window, and the truncated reads shorter than 50 bp were discarded; reads containing ambiguous characters were also discarded. (ii) Only overlapping sequences longer than 10 bp were assembled according to their overlapped sequence. The maximum mismatch ratio of overlap region is 0.2. Reads that could not be assembled were discarded.

### 2.6. Data Processing and Analysis

The cloud platform of Shanghai Majorbio Technology Co., Ltd. (Shanghai, China) was used for interactive cloud analysis of the biological information (http://www.i-sanger.com, accessed on 3 May 2021). UPARSE software (version 7.1 http://drive5.com/uparse/, accessed on 3 May 2021) was used to perform operational taxonomic unit (OTU) clustering and eliminate chimeras based on 97% similarity. Each sequence was annotated for species classification using the RDP classifier (http://rdPh.cme.msu.edu/, accessed on 3 May 2021) and compared with the SILVA and UNITE databases. The OTUs were used to analyze the composition of the microbial community structure, its α-diversity, microbial correlations and microbial function prediction, and then the graphs were created. SPSS 22.0 (IBM, Inc., Armonk, NY, USA) was used for statistical analyses. One-way ANOVA was used to compare the differences of the growth indices of *B. striata* and the diversity of root endophytic fungi under different bamboo forests and different stand densities for each bamboo forest, and a least significant difference (LSD) multiple comparison was used to analyze the significance ($p < 0.05$). All the graphs were plotted in Origin 2018 (OriginLab, Northampton, MA, USA) and Adobe Illustrator CC 2018 (San Jose, CA, USA).

## 3. Results

### 3.1. Effects of Bamboo Forest Type and Density on the Growth of B. striata

The leaf width and dry weight of the aboveground and underground biomass, root length, root width and root number of *B. striata* were significantly higher in the *Phyllostachys edulis* forest than in the *P. iridescens* forest, while those in the *P. iridescens* forest were significantly higher than those in the *P. glauca* forest (Figure 1C–E,G–I). The dry weight ratio of the aboveground to the underground biomass of *B. striata* in the *P. iridescens* forest was significantly the highest, and plant height and leaf length were the largest in the three bamboo forests (Figure 1A,B,F).

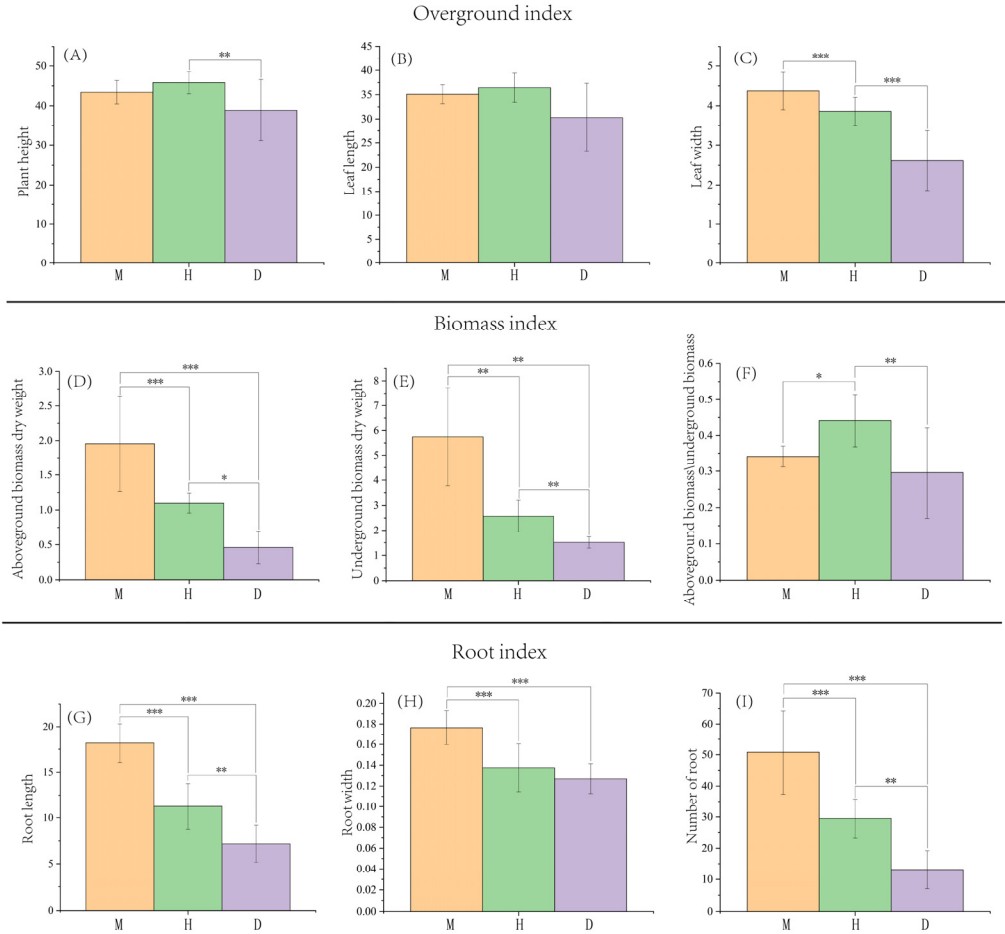

**Figure 1.** Growth indices of *Bletilla striata* under different bamboo forests. (**A**) Plant height; (**B**) Leaf length; (**C**) Leaf width; (**D**) Aboveground biomass dry weight; (**E**) Underground biomass dry weight; (**F**) Aboveground biomass/underground biomass; (**G**) Root length; (**H**) Root width; (**I**) Number of root; M, H, D: *Phyllostachys edulis*, *P. iridescens*, *P. glauca*, respectively. * $p < 0.05$. ** $p < 0.01$. *** $p < 0.001$.

There were also differences in the growth of *B. striata* under different densities of bamboo forests. The leaf width, root width, dry weight of the aboveground and belowground biomass of *B. striata* in *P. edulis* forest were significantly higher in the low-density than in the high-density sections of the forest. The leaf length in the low-density group in *P. iridescens* was significantly higher than that in the high-density group, and the root number and belowground biomass dry weight in the medium density group were significantly higher than those in the high-density group; the dry weight ratio of the aboveground and belowground biomass in the high-density group was significantly higher than that in the low-density group (Table 1).

**Table 1.** Growth indices of *Bletilla striata* under different planting densities (mean ± SD).

| | | Plant Height (cm) | Leaf Length (cm) | Leaf Width (cm) | Root Length (cm) | Root Width (cm) | Number of Roots (num.) | Aboveground Biomass (g) | Belowground Biomass (g) | Aboveground/Belowground Biomass |
|---|---|---|---|---|---|---|---|---|---|---|
| *P. edulis* | high | 44.24 ± 4.97 [a] | 35.51 ± 0.78 [a] | 4.01 ± 0.22 [b] | 17.60 ± 2.38 [a] | 0.16 ± 0.00 [b] | 41.3 ± 4.49 [a] | 1.32 ± 0.18 [b] | 3.6 ± 0.28 [b] | 0.36 ± 0.02 [a] |
| | medium | 42.38 ± 0.08 [a] | 34.39 ± 1.19 [a] | 4.28 ± 0.48 [a b] | 18.44 ± 2.15 [a] | 0.17 ± 0.01 [a b] | 48.83 ± 14.37 [a] | 1.92 ± 0.57 [a b] | 5.9 ± 1.17 [a] | 0.32 ± 0.03 [a] |
| | low | 43.64 ± 2.70 [a] | 35.39 ± 3.49 [a] | 4.84 ± 0.28 [a] | 18.50 ± 2.70 [a] | 0.19 ± 0.02 [a] | 62.27 ± 12.56 [a] | 2.63 ± 0.52 [a] | 7.75 ± 1.14 [a] | 0.34 ± 0.02 [a] |
| *P. iridescens* | high | 45.45 ± 2.93 [a] | 36.12 ± 3.29 [a b] | 3.97 ± 0.44 [a] | 10.14 ± 2.14 [a] | 0.14 ± 0.03 [a] | 22.77 ± 1.00 [b] | 0.97 ± 0.05 [a] | 1.89 ± 0.200 [b] | 0.52 ± 0.07 [a] |
| | medium | 43.89 ± 1.93 [a] | 34.14 ± 1.82 [b] | 3.92 ± 0.48 [a] | 10.01 ± 2.80 [a] | 0.14 ± 0.02 [a] | 33.97 ± 6.69 [a] | 1.18 ± 0.19 [a] | 2.77 ± 0.62 [a] | 0.43 ± 0.03 [a b] |
| | low | 48.18 ± 2.10 [a] | 39.03 ± 1.88 [a] | 3.69 ± 0.16 [a] | 13.62 ± 0.38 [a] | 0.12 ± 0.02 [a] | 31.73 ± 2.29 [a] | 1.15 ± 0.05 [a] | 3.06 ± 0.26 [a] | 0.38 ± 0.02 [b] |
| *P. glauca* | high | 40.53 ± 3.17 [a] | 31.39 ± 2.76 [a] | 2.64 ± 0.38 [a] | 7.08 ± 2.20 [a] | 0.12 ± 0.02 [a] | 13.05 ± 3.27 [a] | 0.5 ± 0.17 [a] | 1.65 ± 0.13 [a] | 0.30 ± 0.08 [a] |
| | medium | 33.68 ± 6.35 [a] | 25.67 ± 5.82 [a] | 2.17 ± 0.60 [a] | 6.61 ± 1.79 [a] | 0.12 ± 0.01 [a] | 9.72 ± 3.28 [a] | 0.31 ± 0.20 [a] | 1.29 ± 0.27 [b] | 0.24 ± 0.12 [a] |
| | low | 42.44 ± 11.30 [a] | 33.85 ± 10.17 [a] | 3.03 ± 1.10 [a] | 7.83 ± 2.65 [a] | 0.13 ± 0.02 [a] | 16.55 ± 9.58 [a] | 0.57 ± 0.31 [a] | 1.61 ± 0.05 [a] | 0.35 ± 0.18 [a] |

Same lowercase letters within a column of the same bamboo forest indicate no significant difference among different densities ($p < 0.05$).

### 3.2. Composition of Endophytic Fungal OTUs in B. striata under Different Bamboo Forest Types and Densities

The classification diagram of endophytic fungal OTUs in *B. striata* roots shows the number of common and unique OTUs in the root samples from three bamboo forests and the OTU comparison among different densities in the same bamboo forest. As shown in Figure 2, the number of endophytic fungal OTUs in *B. striata* under different bamboo species was as high as 421 in the *P. iridescens* forest. There were 80 fungal OTUs shared among the three bamboo forests, while there were 213, 247 and 108 fungal OTUs specific to a single forest of *P. edulis*, *P. iridescens* and *P. glauca*, respectively. There also existed some fungal OTUs shared by two bamboo forests (Figure 2A).

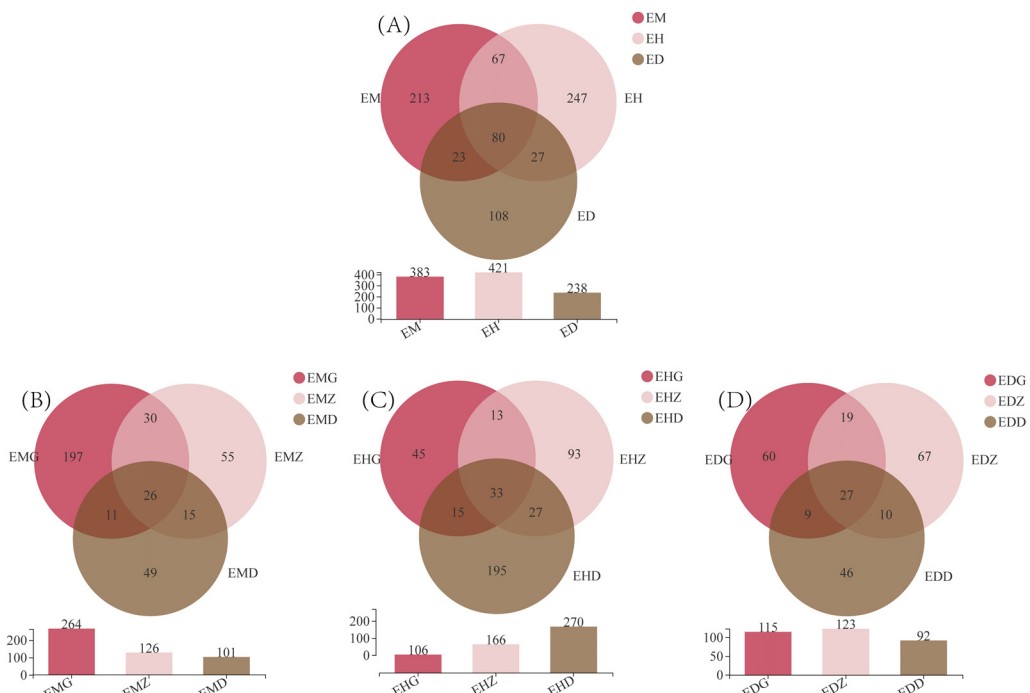

**Figure 2.** Venn diagram of the OTU classification of endophytic fungi in the *Bletilla striata* roots under different planting types and densities: (**A**) three bamboo forest models; (**B**) *Phyllostachys edulis* model; (**C**) *P. iridescens* model; (**D**) *P. glauca* model; EM, EH, ED: *P. edulis*, *P. iridescens*, *P. glauca*, respectively; EMG, EMZ, EMD: *P. edulis* at high-, medium- and low-density, respectively; EHG, EHZ, EHD: *P. iridescens* at high-, medium- and low-density, respectively; EDG, EDZ, EDD: *P. glauca* at high-, medium- and low-density, respectively. OTU, operational taxonomic unit.

The number of OTUs of *B. striata* under different densities of bamboo also differed. The number of OTUs in the high-density group of *P. edulis* was as high as 264, and the number of endemic species was 197, which was significantly higher than that in the medium-density group (all: 126, endemic species: 55) and the-low density group (all: 101, endemic species: 49) (Figure 2B). However, the trend was opposite in the *P. iridescens* forest. The highest number of OTUs was 270 in the low-density group, and 195 were endemic species, which were higher than those in the medium-density group (all: 166, endemic species: 93) and high-density group (all: 106, endemic species: 45) (Figure 2C). The difference in the numbers of OTUs was relatively small under different densities in the *P. glauca* forest. The highest number of OTUs was 123 in the medium-density group (Figure 2D).

### 3.3. Diversity Analysis of the Endophytic Fungal Community in B. striata

At the level of OTUs, the results of the sample coverage index showed that the coverage rate in the collected samples was 99.977 to 99.984% (Figure 3C). These results indicated that the microbial information in each sample was fully measured, and the results represented

the actual situation of endophytic fungi in the *B. striata* roots. The α-diversity index of the endophytic fungi in the *B. striata* roots from three bamboo forests did not differ significantly, but all showed that *P. iridescens* > *P. edulis* > *P. glauca* (Figure 3A,B).

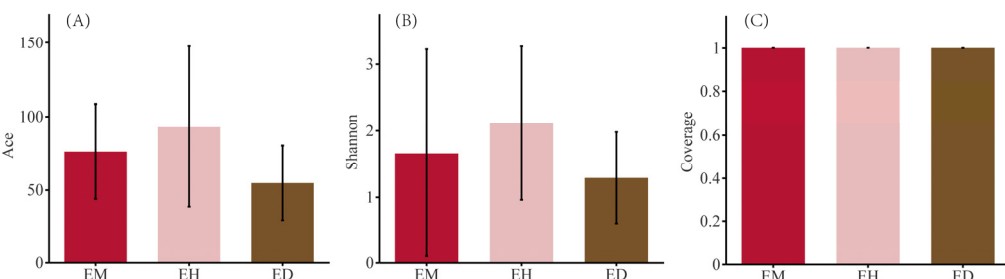

**Figure 3.** Analysis of the α-diversity of endophytic fungi in the *Bletilla striata* roots at OTU level under different bamboo forests: EM, *Phyllostachys edulis* model; EH, *P. iridescens* model; ED, *P. glauca* model. (**A**) Ace; (**B**) Shannon; (**C**) Coverage.

Compared with the bamboo forest type, the stand density had a stronger influence on the diversity and richness of endophytic fungi in the *B. striata* roots. The diversity index in the *P. edulis* high-density group was 3.32, which was significantly higher than that in the low-density group (0.39), and the richness index was the largest in the high-density group. However, the difference was not significant (Figure 4A,D). Among the three density groups of the *P. iridescens* forest, the richness index of the low-density group was significantly the highest (159.7), followed by the medium-density group (69.9) and the high-density group (48.16) (Figure 4B). There was no significant difference in the richness and diversity index of the *P. glauca* forest, which were both relatively low in the low-density group (Figure 4C,F).

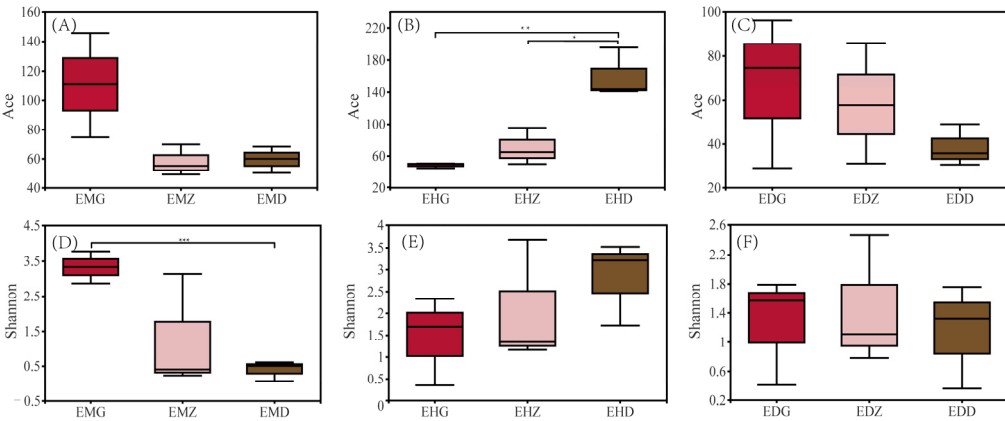

**Figure 4.** Analysis of the α-diversity of the endophytic fungi in *Bletilla striata* roots at OTU level at different densities: EMG, EMZ, EMD, *Phyllostachys edulis* at high-, medium- and low-density, respectively; EHG, EHZ, EHD, *P. iridescens* at high-, medium- and low-density, respectively; EDG, EDZ, EDD, *P. glauca* at high-, medium- and low-density, respectively. (**A**–**C**), Ace for *P. edulis*, *P. iridescens* and *P. glauca*, respectively; (**D**–**F**), Shannon for *P. edulis*, *P. iridescens* and *P. glauca*, respectively. * $p < 0.05$. ** $p < 0.01$. *** $p < 0.001$.

### 3.4. Analysis of the Community Composition of Endophytic Fungi in B. striata Roots

The community composition of endophytic fungi in *B. striata* roots under different bamboo forest types and densities was studied by high-throughput sequencing technology. As indicated in Figure 5, the endophytic fungi (relative abundance > 0.01) in the roots of *B. striata* in the bamboo forests included five phyla. Those in which the phyla were dominated by Basidiomycota and Ascomycota comprised the highest proportion (Figure 5A).

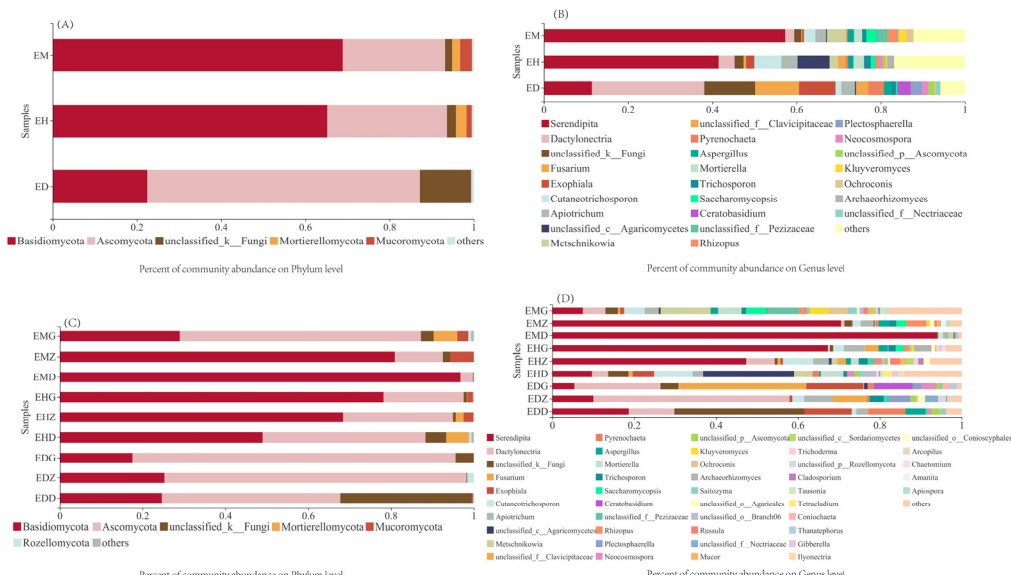

**Figure 5.** Abundance of the endophytic fungi from *Bletilla striata* roots in different grouping patterns. Community composition of the endophytic fungi from *B. striata* roots of three bamboo forests at the phylum (**A**) and genus (**B**) levels. Community composition of endophytic fungi of the *B. striata* roots at the phylum (**C**) and genus (**D**) levels at different densities: EM, EH, ED, *Phyllostachys edulis, P. iridescens, P. glauca*, respectively; EMG, EMZ, EMD, *P. edulis* at high-, medium- and low-density, respectively; EHG, EHZ, EHD, *P. iridescens* at the high-, medium- and low-density, respectively; EDG, EDZ, EDD, *P. glauca* at the high-, medium- and low-density, respectively.

The relative abundance of *B. striata* fungi differed significantly among different types of bamboo forests. Basidiomycota was the dominant phylum of endophytic fungi in the *P. edulis* and *P. iridescens* forests, comprising 68.87% and 65.18%, respectively, while Ascomycota comprised 24.31% and 28.43%, respectively. The opposite was true in the *P. glauca* forest. Ascomycota was the dominant phylum that comprised 64.72%, while Basidiomycota comprised 22.47% (Figure 5A). The abundance of endophytic fungi in the *B. striata* roots also clearly changed under different densities. The abundance of Basidiomycota increased with the decrease in density of *P. edulis* forests, comprising 28.95% in the high-density group and 96.74% in the low-density group. The opposite was true in the *P. iridescens* forest. The abundance of Basidiomycota decreased with the decrease in bamboo density, comprising 78.18% in the high-density group and 48.96% in the low-density group. The overall abundance of Basidiomycota was low in the *P. glauca* forest, and the variation in the abundance was small among the three density groups. In addition, the abundance of Ascomycota at different densities changed in response to changes in Basidiomycota (Figure 5C).

At the genus level, *Serendipita* was the dominant genus of endophytic fungi in *B. striata* in the *P. edulis* and *P. iridescens* forests, comprising 57.30% and 41.46%, respectively. The community composition of endophytic fungi in *B. striata* in the *P. glauca* forest tended to be complex. The proportion of *Dactylonectria* increased to 26.70%, while that of *Serendipita* was only 11.38% (Figure 5B). The dominant genera of endophytic fungi in *B. striata* in the *P. edulis* and *P. iridescens* forests varied substantially under different densities of bamboo forests. The abundance of *Serendipita* increased as the density in *P. edulis* forest decreased, comprising 7.48% in the high-density group and 93.91% in the low-density group. The opposite was true in the *P. iridescens* forest. The abundance of *Serendipita* decreased with decreasing density, comprising 67.30% in the high-density group and 9.71% in the low-density group. *Dactylonectria* was the dominant species in the *P. glauca* forest, comprising the highest proportion (47.92%) in the medium-density group. The abundance of *Serendipita* changed similarly in the *P. edulis* forest, comprising 5.40% in high-density and 18.69% in low-density conditions (Figure 5D).

### 3.5. LefSe Difference Analysis of Endophytic Fungi Community in the B. striata Roots

A linear discriminant analysis effect size (LefSe) was used to study the groups with significant effects on the diversity of abundance under different bamboo forests and densities. As shown in Figure 6, the endophytic fungi in the *B. striata* roots of three bamboo forests differed significantly. There were 11 different indicator species in the *P. edulis* forest, including Basidiomycota and Chytridiomycota. The rest included g_*Serendipita* (f_Serendipitaceae, o_Sebacinales order to genus), c_Agaricomycetes, g_Metschnikowia (f_Metschnikowiacea, o_Saccharomycetales, c_Saccharomycetes class to genus) and f_Dipodascaceae (o_Saccharo mycetales, c_Saccharomycetes class to family). There were nine different indicator species in the *P. iridescens* forest, including g_*Cutaneotrichosporon*, g_*Candida* (f_Saccharomycetales_fam _Incertae_sedis family to genus), g_*Acremonium*, g_*Penicillium*, g_*Cyberlindnera* (f_Phaffomy cetaceae family to genus), g_*Chaetomium* and g_*Trichosporon*. There were three different indicator species in the *P. glauca* forest, including o_Hypocreales (c_Sordariomycetes, p_Ascomycota phylum to order) (Figure 6A).

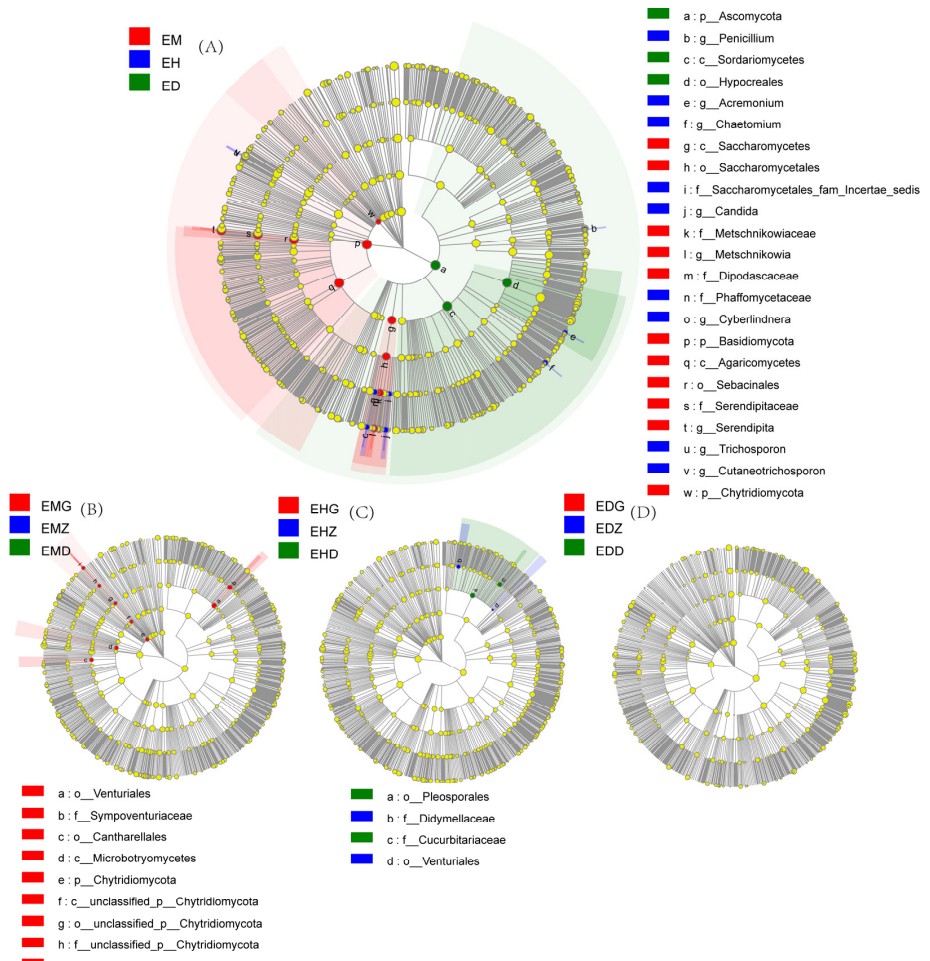

**Figure 6.** LefSe discriminant diagram of the endophytic fungi communities of *Bletilla striata* under different bamboo forest types and densities: (**A**) three bamboo forest models; (**B**) *Phyllostachys edulis* model; (**C**) *P. iridescens* model; (**D**) *P. glauca* model. Different color nodes indicate the groups that are enriched in the corresponding groups and have a significant influence on the differences between groups. The yellow nodes represent the microbial groups with no significant difference among the different groups, and the node diameter represents the species abundance: EM, EH, ED, *P. edulis*, *P. iridescens*, *P. glauca*, respectively; EMG, EMZ, EMD, *P. edulis* at high-, medium- and low-density, respectively; EHG, EHZ, EHD, *P. iridescens* at high-, medium- and low-density, respectively; EDG, EDZ, EDD, *P. glauca* at high-, medium- and low-density, respectively; LEfSe, linear discriminant analysis effect size.

Under the three densities, the different species in the high-density of the *P. edulis* forest were f_Sympoventuriaceae (o_Venturiales order to family), o_Cantharellales and c_Microbotryomycetes (Figure 6B). F_Didymellaceae and o_Venturiale were two distinct species in the medium-density forest of *P. iridescens*, F_Cucurbitariaceae (o_Pleosporales order) was found in the low-density forest (Figure 6C). There were no different species in the three densities of *P. glauca* (Figure 6D).

*3.6. Correlation Analysis of Environmental Factors and the Endophytic Fungi Community in B. striata Roots*

Collinearity analysis was performed on 18 environmental factors, including the pH, EC, OM, TN, AN, NN, TP, AP, TK, AK, MBC, MC, SBD, SOC, C:N, C:P, N:P and LI, and 5 soil indices, including TN, NN, TP, SOC and C:P. Indices that had variance influence factor (VIF) values > 10 were removed, and then the redundancy analysis (RDA) was started. The results showed that environmental factors explained the changes in endophytic fungi community in 71.87% of *B. striata* at the phylum level, with TK, AP, LI, SBD, pH and MC positively correlated with Basidiomycota and negatively correlated with Ascomycota. C:N, EC, OM, N:P, AK, AN and MBC positively correlated with Ascomycota and negatively correlated with Basidiomycota (Figure 7A). The SBD ($p = 0.001$, $R^2 = 0.57$) and LI ($p = 0.002$, $R^2 = 0.33$) had significant effects on the community composition of endophytic fungi in *B. striata*. N:P ($p = 0.05$, $R^2 = 0.24$), MBC ($p = 0.08$, $R^2 = 0.19$) and OM ($p = 0.07$, $R^2 = 0.22$) also had some influence on the community composition of endophytic fungi in *B. striata* (Table 2).

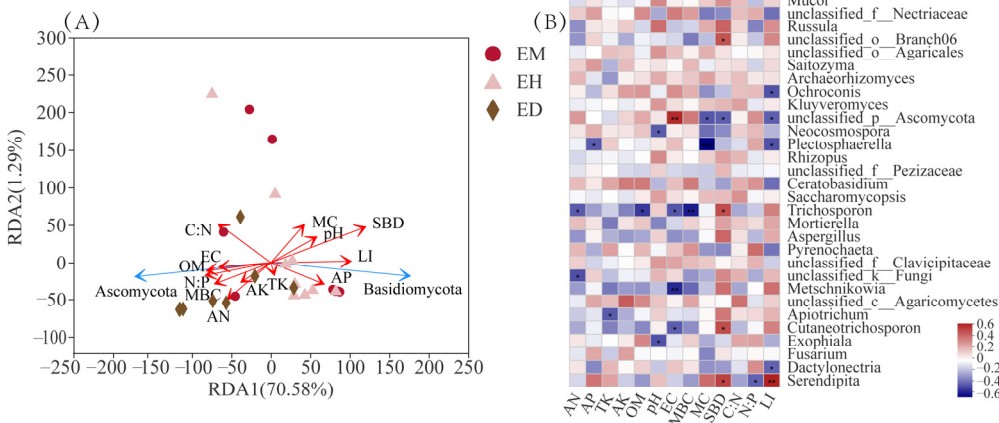

**Figure 7.** Correlation analysis of the endophytic fungi and environmental factors in *Bletilla striata* roots of different bamboo forests: (**A**) RDA chart at the phylum level; (**B**) heatmap analysis chart at the genus level: EM, *Phyllostachys edulis*; EH, *P. iridescens*; ED, *P. glauca*; RDA, redundancy analysis. The blue and red arrows represent endophytic fungi and environmental factors, respectively.

**Table 2.** Correlation data table of RDA analysis between environmental factors and endophytic fungi of *Bletilla striata* roots.

|  |  | AN | AP | TK | AK | OM | Ph | EC | MBC | MC | SBD | C:N | N:P | LI |
|---|---|---|---|---|---|---|---|---|---|---|---|---|---|---|
| Endophytic | $p$ | 0.13 | 0.18 | 0.96 | 0.48 | 0.07 | 0.17 | 0.19 | 0.08 | 0.25 | 0.00 | 0.12 | 0.05 | 0.02 |
| fungi | $R^2$ | 0.17 | 0.14 | 0.00 | 0.06 | 0.22 | 0.14 | 0.14 | 0.19 | 0.12 | 0.57 | 0.17 | 0.24 | 0.33 |

$p < 0.05$ represents a significant influence; $R^2$, higher values indicate a closer association.

The effects of different environmental factors on the endophytic fungal community of *B. striata* roots could be observed from the heatmap. In the *P. iridescens* and *P. edulis* forests, *Serendipita* significantly positively correlated with the SBD and LI and negatively correlated with N:P. The dominant genus *Dactylonectria* significantly negatively correlated with LI in the *P. glauca* forest (Figure 7B).

*3.7. FUNGuild Function Prediction Analysis of the Endophytic Fungi in B. striata Roots*

Based on the FUNGuild function prediction, endophytic fungi in the roots of *B. striata* were divided into nine categories, including Undefined, Pathogen–Saprotroph–Symbiotroph, Pathotroph, Pathotroph–Saprotroph, Pathotroph–Saprotroph–Symbiotroph, Pathotroph–Symbiotroph, Saprotroph, Saprotroph–Symbiotroph and Symbiotroph. Symbiotrophs comprised 59% of the *B. striata* roots in *P. edulis*, 43% in *P. iridescens*, and 12% in *P. glauca*. Correspondingly, the proportion of saprotrophic fungi in the *P. glauca* forest increased significantly, comprising 42%, followed by the *P. iridescens* forest (24%), and the *P. edulis* forest (23%) (Figure 8A).

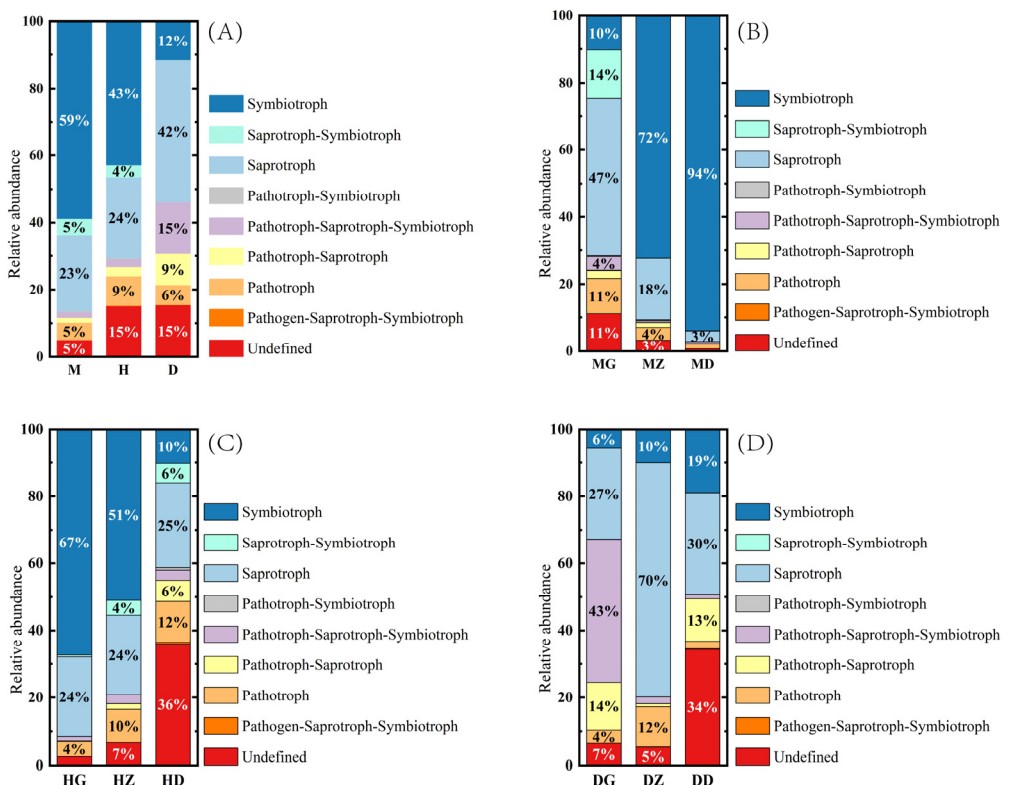

**Figure 8.** FUNGuild function prediction of the endophytic fungi of *Bletilla striata* in different grouping models: (**A**) three bamboo forest models; (**B**) *Phyllostachys edulis* model; (**C**) *P. iridescens* model; (**D**) *P. glauca* model. M, H, D, *P. edulis, P. iridescens, P. glauca*, respectively; MG, MZ, MD, *P. edulis* at high-, medium- and low-density, respectively; HG, HZ, HD, *P. iridescens* at high-, medium- and low-density, respectively; DG, DZ, DD, *P. glauca* at high-, medium- and low-density, respectively.

In addition, the density of bamboo forest had a significant effect on the nutrient type of endophytic fungi in the B. striata roots. In the *P. edulis* forest, saprotrophs (47%) comprised the highest proportion in the high-density group, while symbiotrophs comprised only 10%. However, symbiotrophs comprised 72% and 94% in the medium- and high-density groups, respectively, which were absolutely dominant (Figure 8B). Saprotrophs and pathotrophs were the primary nutrient types in the *P. glauca* forest. Saprotrophs comprised 70% of the medium-density group, and Pathotroph–Saprotroph–Symbiotroph comprised 43% of the high-density group (Figure 8D). In the *P. iridescens* forest, symbiotrophs and saprotrophs were the primary types of nutrient consumers, but their changing trends differ from those of the other two bamboo forests. The symbiotrophic fungi were 67% at high-density and 10% at low-density. Saprotrophs comprised 24–25% at all three densities (Figure 8C).

## 4. Discussion

### 4.1. Effects of Different Bamboo Forest Types and Densities on the Growth of B. striata

In this study, bamboo forest type and stand density had significant influences on the growth of *B. striata*. The growth of the aboveground and belowground parts of *B. striata* in the *P. edulis* stand was clearly superior among the three bamboo stands, and the dry matter accumulation of roots, tubers, stems and leaves was significantly higher ($p < 0.05$), indicating that the environment under *P. edulis* forest is the most suitable for the growth of *B. striata*. The plant height and leaf lengths of *B. striata* in the *P. iridescens* forest have some advantages, and the dry weight ratio of the aboveground and belowground biomass was significantly higher than that of the other bamboo forests, indicating that the growth of *B. striata* in the *P. iridescens* forest is more focused on the aboveground than the belowground parts, which is similar to the results of a study that found that the biomass distribution of plants under low light conditions is more directed to the aboveground part [33,34]. The soil nutrient indices, such as N, K, OM and MBC, were all superior in the *P. glauca* forest, but the growth of *B. striata* was significantly inhibited compared with the other bamboo forests. This could be owing to the extreme forest canopy density and the complexity of the root microbial composition in *B. striata*. Saprophytic and pathogenic fungi, such as *Dactylonectria*, increased. In addition, the leaf width of *B. striata* in the *P. iridescens* and *P. glauca* forests decreased significantly, because under high canopy density, the plants would sacrifice part of their photosynthetic processes to form narrower, thinner or heavier leaves, thus, increasing the density of leaf tissue [35]. Observing the growth of *B. striata* under different densities of the same bamboo forest enabled the observation that the lower density of bamboo forest has clear advantages to the growth of *B. striata*. Studies have shown that *B. striata* can grow normally under a canopy density of 0.2~0.9, but the optimal shade is 45~55% [36,37]. Bamboo forests usually have a high canopy density. In this study, the average LI in the *P. edulis* forest decreased by 76.53% compared with the LI outside of the company, and the LI in the low-density group decreased the least (52.72%). Therefore, low-density *P. edulis* forests affect the growth of *B. striata* by best meeting its demand for light.

### 4.2. Effects of Different Bamboo Forest Types and Densities on the Endophytic Fungal Community in B. striata Roots

Plant growth is closely related to the colonization of fungi, and the colonization of endophytic fungi in plant roots is also affected by its environment. This study found that the change in bamboo species and density affected the abundance and diversity of endophytic fungi in the *B. striata* roots. In comparison, the abundance and diversity of endophytic fungi in the *B. striata* roots were the highest in *P. iridescens* forests, indicating that the environment under the *P. iridescens* forest may be favorable to the growth of endophytic fungi in the roots of *B. striata*. However, the effect on *B. striata* is double-sided, which could be caused by the decrease in mycorrhizal fungi and the increase in pathogenic fungi [38]. This is consistent with the results of FUNGuild functional prediction analysis in this study.

In this study, Basidiomycota and Ascomycota were the dominant fungi in the roots of *B. striata*, and the composition of their abundance changed significantly under different bamboo forest types and densities, indicating that the external environment would significantly affect the endophytic fungal community composition of *B. striata*, which was similar to the results of previous studies [39,40]. Simultaneously, these changes had a clear correlation with the nutrient types of endophytic fungi in *B. striata* roots. There were 45 genera of endophytic fungi with relative abundance greater than 1% in the *Bletilla striata* roots, including nine nutrient types. Symbiotrophic, saprotrophic and pathotrophic fungi differed in their abundance in the three bamboo forests. This difference effectively reflected the different selection of endophytic fungi in the *B. striata* roots when grown under different bamboo forests. In addition, 10 genera of Orchidaceae mycorrhizal fungi have been reported in three bamboo forests, including *Serendipita, Fusarium, Pyrenochaeta, Aspergillus, Ceratobasidium, Neocosmospora, Russula, Trichoderma, Thanatephorus,* and *Chaetomium* [41,42].

The LEfSe method was used to analyze the different indicator species of the composition of the endophytic fungi community in the *B. striata* roots in different groups, which could reflect the selection results of endophytic fungi in bamboo forests to some extent. Symbiotic fungi were the primary endophytic fungi in *B. striata* in the *P. edulis* forests. *Serendipita* ranked the first, and it is a mycorrhizal fungus of the orchid family. Pathogenic and saprophytic fungi increased in the *P. iridescens* forest, but there were still symbiotic fungi and orchid mycorrhizal fungi such as *Penicillium* and *Chaetomium*. The proportion of saprophytic fungi increased substantially in the *P. glauca* forest, and almost no symbiotic fungi were found. Thus, the *P. edulis* forest was the most suitable environment for the growth of symbiotic fungi in *B. striata*. In addition, there were more saprophytic and pathogenic differential species in groups in bamboo forests with higher densities. The results showed that the low-density environment was suitable for symbiotic fungal colonization, and thus, the growth of *B. striata* under the bamboo forest was affected.

*4.3. Potential Effects of Serendipita on the Growth of B. striata*

*Serendipita* was the most abundant endophytic fungus in the *B. striata* roots in the bamboo forest. It is a member of the Serendipitaceae family in Basidiomycota, is widely distributed in nature and can form highly diversified root symbionts with many plants. Studies have shown that the root system of host plants was often significantly increased by artificial inoculation [43,44], which was consistent with the results of this study that *Serendipita* was the most abundant in *P. edulis* forests where the root length, root width and root numbers of *B. striata* were significantly the highest. This result was the most pronounced in *P. edulis* low-density forests. In addition, the abundance of *Serendipita* positively correlated with most of the growth indices of *B. striata*. This is probably because the Serendipitaceae have distinctive molecular mechanisms of action on host plants. It can promote plant growth and enhance plant resistance to abiotic and pathogen stresses [45].

The abundance of *Serendipita* displayed obvious gradient changes in different bamboo forest groups, suggesting that the bamboo forest environment could significantly change the abundance of *Serendipita*. A correlation analysis showed that the abundance of *Serendipita* on *B. striata* grown under bamboo was significantly affected by the SBD, LI and N:P, which was consistent with previous studies on the correlation between the abundance of *Sebacinales* fungi and elements such as N and pH. Simultaneously, *Sebacinales* mycorrhizae often exist in barren soils, such as those in southwestern Australia, or in plants that thrive in severe soils and exhibit some drought resistance [43]. Therefore, *Serendipita* can be used to inoculate orchids in unfavorable soils to produce green applications to improve the competitiveness and stress resistance of plants, expand the planting range of plants, and provide more effective guidance for the introduction, cultivation, production and sustainable development of orchids.

**5. Conclusions**

The growth and development of *B. striata* were significantly affected by the types and densities of bamboo forests. The environment in the *Phyllostachys edulis* forest was the most favorable for the whole growth of *B. striata* and the best under low density, and the primary growth of *B. striata* in the *P. iridescens* forest was in the aboveground parts. The community composition and diversity of endophytic fungi in the *B. striata* roots differed under different bamboo forest types and densities. Basidiomycota was the main endophytic fungi in the *P. edulis* and *P. iridescens* forests, and symbiotrophic fungi were the most dominant type based on the analysis of nutrient usage. Ascomycota was the dominant phylum in the *P. glauca* forest, and saprotrophic fungi were the main nutrient types. The soil bulk density and light intensity were the primary factors that affected the abundance of these two phyla. The dominant species of endophytic fungi in the *B. striata* roots also differed in different types of bamboo forests. *Serendipita* was the dominant genus in the *P. edulis* and *P. iridescens* forests, and *Dactylonectria* was the dominant species in the *P. glauca* forest. The abundances of Basidiomycota and *Serendipita* increased with the increase in bamboo density in the

*P. edulis* forest but showed an opposite trend in *P. iridescens* forests. In this study, *Serendipita* was the primary mycorrhizal fungi of *B. striata* in three bamboo forests, and most growth indices positively correlated with *Serendipita*, which was greatly affected by the density of bamboo. Therefore, in the application and production of afforestation, we can select suitable strains based on site conditions and prepare fungal agents to inoculate plants to improve their adaptability and stress resistance. This study provides new ideas to deepen the study of plant configuration under bamboo forests and the application of growing orchids under these forests.

**Author Contributions:** Conceptualization, W.H. and X.Z.; methodology, H.F. and X.Z.; software, H.F. and X.Z.; validation, W.H. and S.L. (Siren Lan); formal analysis, H.F.; investigation, H.F and Z.S.; resources, Z.S. and S.L. (Shanmin Li); data curation, X.Z.; writing—original draft preparation, H.F.; writing—review and editing, H.F. and X.Z.; visualization, H.F.; supervision, W.H., S.L. (Siren Lan) and X.Z.; project administration, W.H.; funding acquisition, W.H. and X.Z. All authors have read and agreed to the published version of the manuscript.

**Funding:** This research was funded by Shanghai Agricultural Development Project of Science and Technology (2021-02-08-00-12-F00778), National Natural Science Foundation of China (31902108), and the Special Fund for Scientific Research of Shanghai Landscaping & City Appearance Administrative Bureau (G222406).

**Institutional Review Board Statement:** Not applicable.

**Data Availability Statement:** Data are available upon request from the corresponding author.

**Acknowledgments:** We acknowledge and thank all the teachers and classmates from Shanghai Chenshan Botanical Garden and Fujian Agriculture and Forestry University who offer help in the surveys.

**Conflicts of Interest:** The authors declare no conflict of interest. The funders had no role in the design of the study; in the collection, analyses, or interpretation of data; in the writing of the manuscript, or in the decision to publish the results.

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
