# Peer review of "Effects of Bamboo Forest Type and Density on the Growth of Bletilla striata and Root Endophytic Fungi"

_diversity, doi:10.3390/d14050391_

Round 1

Reviewer 1 Report

For overall, this manuscript is well written with a few corrections.

Author Response

Dear Reviewer,

Thank you very much for your comments concerning our manuscript entitled “Effects of bamboo forest type and density on the growth of Bletilla striata and root endophytic fungi” (ID: 1695884). These comments are all valuable and helpful for revising and improving our manuscript. We have studied the comments carefully and have made correction which we hope to meet with approval. Revised portions are marked in red in the present revised manuscript.

We appreciate for your warm work earnestly, and hope that the corrections will meet with approval.

Once again, thank you very much for your comments and suggestions.

We look forward to your information about our revised paper and thank you for your good comments.   

Yours sincerely,

Xinhua Zeng

Shanghai Chenshan Plant Science Research Center, Chinese Academy of Sciences, Chenshan Botanical Garden

No. 3888, Chenhua Road, Shanghai 201602, China

E-mail: zengxinhua@csnbgsh. cn

Tel: +008617721190554

Reviewer 2 Report

Paper deals with bamboo forest type and density effect on Bletilla orchids and mycorrhiza and certainly has importance in this field. In all three studied bamboo species there were low, medium and high density plots. My main concern is that the high density in P.edulis was still lower than low densities of two other species. You should explain much better how these densities are comparable.

Minor remarks:

line 51-52 genera names should be with capital letter

line 66 also peninsula name is with capital letter

line 84 nutrients and soil should not be with capital letter

line 92 B. striata should be in italic

line 100-102 grammatically wrong

Reviewer 3 Report

Fu et al. manuscript report on the effects of bamboo forest type and density on the growth of Bletilla striata and root endophytic fungi. The research design comprising bamboo forest type and density is appropriate to respond to the stated questions. In general, the manuscript is well written and flows logically. However, some sections must be improved before publication. In the introduction, cite previous studies describing Bletilla striata endophytes (e.g. Zeng et al. 2021). In methods section, please describe how forward and reverse reads were processed, particularly the filtering process. In results, in the analysis of the community composition title, consider than relative abundance is based on reads. How unequal sequencing depth among simples was handled? Relative abundance is important, but why are the results of the community composition not presented according to the richness of OTUs? Finally, in the discussion section, consider the effect that primer selection has on the results. There are many references on the subject and the most used region is ITS2 (e.g. Waud et al. 2014).

Waud, M., Busschaert, P., Ruyters, S., Jacquemyn, H., and Lievens, B. (2014). Impact of primer choice on characterization of orchid mycorrhizal communities using 454 pyrosequencing. Mol. Ecol. Resour. 14, 679–699. doi: 10.1111/1755-0998.12229

Zeng, X.; Diao, H.; Ni, Z.; Shao, L.; Jiang, K.; Hu, C.; Huang, Q.; Huang, W. Temporal Variation in Community Composition of Root Associated Endophytic Fungi and Carbon and Nitrogen Stable Isotope Abundance in Two Bletillaspecies (Orchidaceae). Plants 2021, 10, 18. https://doi.org/10.3390/plants10010018

Specific comments

L 35-36 Please clarify the concept “the internal ecological environment of the system”

L 120-121 Please clarify the source of B. striata plants (in-vitro or greenhouse). Were the roots of the plants free of fungal colonization?

L162-163 specific primers for … Please clarify the text.

L214 “The classification diagram of endophytic fungi of the OTUs in B. striata roots shows” please rephrase for “The classification diagram of endophytic fungal OTUs in B. striata roots shows”

L217 Please correct “Figure 1” must be Figure 2

L218-219 “The number of different endemic OTUs in these bamboo forests was clearly higher than that of the common species of bamboo” Please move to Discussion, but you need to contrast this information.

L252-253 Are you referring to the differences? Please clarify the text

Round 2

Reviewer 2 Report

The paper could be accepted

Reviewer 3 Report

The authors have done a great job to incorporate the comments and suggestions.  I think the manuscript is suitable for publication.